# Radiation Effects on Fiber Bragg Gratings: Vulnerability and Hardening Studies

**DOI:** 10.3390/s22218175

**Published:** 2022-10-25

**Authors:** Adriana Morana, Emmanuel Marin, Laurent Lablonde, Thomas Blanchet, Thierry Robin, Guy Cheymol, Guillaume Laffont, Aziz Boukenter, Youcef Ouerdane, Sylvain Girard

**Affiliations:** 1UJM, CNRS, IOGS, Laboratoire Hubert Curien, University of Lyon, UMR 5516, 18 rue Prof. B. Lauras, F-42000 Saint-Etienne, France; 2iXblue Photonics, F-22300 Lannion, France; 3CEA List, Université Paris-Saclay, F-91120 Palaiseau, France; 4CEA, Service d’Études Analytiques et de Réactivité des Surfaces, Université Paris-Saclay, F-91191 Gif-sur-Yvette, France

**Keywords:** fiber gratings, fiber Bragg gratings, fiber sensors, optical fibers, radiation effects, harsh environments

## Abstract

Fiber Bragg gratings (FBGs) are point optical fiber sensors that allow the monitoring of a diversity of environmental parameters, e.g., temperature or strain. Several research groups have studied radiation effects on the grating response, as they are implemented in harsh environments: high energy physics, space, and nuclear facilities. We report here the advances made to date in studies regarding the vulnerability and hardening of this sensor under radiation. First, we introduce its principle of operation. Second, the different grating inscription techniques are briefly illustrated as well as the differences among the various types. Then, we focus on the radiation effects induced on different FBGs. Radiation induces a shift in their Bragg wavelengths, which is a property serving to measure environmental parameters. This radiation-induced Bragg wavelength shift (RI-BWS) leads to a measurement error, whose amplitude and kinetics depend on many parameters: inscription conditions, fiber type, pre- or post-treatments, and irradiation conditions (nature, dose, dose rate, and temperature). Indeed, the radiation hardness of an FBG is not directly related to that of the fiber where it has been photo-inscribed by a laser. We review the influence of all these parameters and discuss how it is possible to manufacture FBGs with limited RI-BWS, opening the way to their implementation in radiation-rich environments.

## 1. Introduction

Silica-based optical fibers (OFs) have attracted a lot of interest from research groups thanks to their properties, such as light weight, small volume, high bandwidth, and resistance to most electromagnetic perturbations, for both telecommunication and sensing applications [1,2]. Among the optical fiber sensors (OFSs) is the fiber Bragg grating (FBG) [3]. It is a punctual sensor, with a length of a few mm to a few cm, photo-inscribed with a laser beam inside the core of a single-mode optical fiber. It is also possible to write FBGs in multimode fibers. However, in this case, the gratings will give rise to several Bragg peaks, as studied for the first time by Wanser et al. in 1994 [4]. In the FBGs, the information is “wavelength encoded”; indeed, each grating causes a dip or peak in the fiber transmission or reflection spectra, respectively, whose peak position is known as the “Bragg wavelength”. The Bragg wavelength depends on the selected FBG characteristics and it spectrally shifts when the environmental parameters around the fiber, such as temperature, strain, pressure, and humidity, evolve. This property makes the FBG a good sensor (From the same family of the FBGs, there is another OFS type: long period grating, or briefly, LPG. Recently, a review about the radiation effects on LPGs has been published [5]; consequently, this subject will not be addressed in this article) as, after calibration, the real-time monitoring of the Bragg wavelength allows the determination of the changes in these measurands (see review [6]).

This sensor type is easily multiplexable; indeed, we can easily write FBGs at different places along the same optical fiber, under the condition that each of them could be distinguished by using wavelength division multiplexing (WDM) or time division multiplexing (TDM) [7]. However, WDM is easier to be employed, since several FBGs can be written in series and investigated with a broadband light source. The only conditions are that each grating must have a different Bragg wavelength and that the spectral ranges of variation of the FBGs must not overlap. This can be achieved by tailoring the FBG characteristics at the manufacturing stage. Typically, about ten FBGs can be implemented using this approach in one fiber sample: this number depends on the spectral range of the acquisition system. TDM, instead, can identify a series of several FBGs, characterized initially by identical (or near) Bragg wavelength values, by combining a pulsed tunable laser in the wavelength sensing range of the gratings and a photodiode with a very high acquisition speed. The two techniques WDM and TDM can be combined, opening the way to quasi-distributed measurements with thousands of FBGs along one fiber (see [8] as an example).

Moreover, compared to other OFSs, such as distributed OFS [9], FBGs present a fast response, opening the way to monitor fast dynamics: the acquisition rate of recent interrogation systems can reach 100 MHz [10]. Such high acquisition rate systems are fundamental to developing vibrational (see review [11]) or ultrasonic sensors [12].

Since the information is wavelength encoded, the FBGs have been largely investigated for their applications in harsh environments, combining radiation and sometimes extreme temperatures [13]. Examples of harsh environments are space, nuclear power plants, and nuclear waste storage, reactor dismantling, and high-energy physics facilities [14]. Each of them is characterized by different temperature ranges and irradiation conditions, such as the nature of particles, fluence (dose), and flux (dose rate) (The dose is the quantity of energy deposited inside in the material of interest, here silica. 1 Gy(SiO_2_) = 1 J/kg. The dose rate is the deposition speed of the energy, expressed in Gy/s. In some articles, another older dose unit is used, the rad: 1 Gy = 100 rad). For example, space is characterized by a low dose rate and low doses (lower than 10^−3^ Gy/h and 10 kGy, respectively) but large temperature variations (between −200 °C and 300 °C). Fusion-devoted facilities, such as the Laser Megajoule (LMJ, CEA, France) or the National Ignition Facility (NIF, Lawrence Livermore National Laboratory, California), instead, are characterized by low doses (less than 1 kGy) and a very high dose rate (up to MGy/s) but are operating at room temperature (RT), whereas the nuclear reactor core is associated with very high doses (up to GGy) and high temperature (up to 800 °C) (see references in [14]).

Radiation can generate point defects, sometimes called color centers, inside the pure or doped silica matrix of the fiber core and cladding by ionization or displacement damage [15]. At very high total ionizing doses or high neutron fluences, densification is also observed [16]. These two radiation effects will degrade the optical fiber and then the fiber grating performances. For example, each induced defect is characterized by its own absorption bands that degrade the optical fiber transmission. Concerning the grating, because of the radiation, the Bragg peak amplitude can be reduced and its position, the Bragg wavelength, can shift, inducing an error in the measurement parameters. The grating radiation response depends on several parameters, such as the inscription conditions, the fiber, the treatments performed before or after the FBG inscription, and the irradiation conditions (nature, dose, dose rate, and temperature). More than one hundred papers have been published about the FBG response to radiation. A complete review of radiation responses of FBGs was published in 2013 by Gusarov and Hoeffgen [13], whereas in [17] the grating response in radiation-free harsh environments was reviewed. Since then, the research has advanced considerably, pushed by new manufacturing techniques for FBGs and by new needs, in particular at very high temperatures. The aim of this paper is to highlight the different parameters that influence the radiation vulnerability of the different grating types and to explain how radiation affects their performance in real applications involving harsh environments. For this purpose, Table 1 reports the main characteristics of radiation environments with a list of references reporting the results of tests carried out in these environments. 

## 2. Operating Principle

An FBG consists of a periodical structuration of the fiber core refractive index, induced by its exposure to laser light (generally on a short fiber length from a few mm to a few cm) [41], as reported in Figure 1. The photo-inscribed grating allows light transmission at all wavelengths except within a small spectral range, for which the light is reflected by the grating. The dip (or peak) in the transmission (or reflection) spectrum, shown in Figure 1, is centered at the wavelength, known as the Bragg wavelength ($\lambda_{B})$ and defined as:(1)$$\lambda_{B}=\frac{2\times n_{\mathrm{eff}}\times \Lambda }{m}$$
where $n_{\mathrm{eff}}$ is the effective refractive index of the fundamental mode measured at $\lambda_{B}$, Λ is the grating period, and m is an integer, indicating the grating or diffraction order. ${\Delta n}^{\mathrm{mod}}$ is the refractive index modulation amplitude, defined as the difference between the refractive index of the zones that are illuminated and the ones that are not. 

Because of the dependence of the effective refractive index and the grating period on the temperature (respectively, due to the thermo-optic effect [42] and to the thermal expansion [43]) and on the strain applied on the fiber (due to the elasto-optic properties [44]), the FBG can be used as a temperature and/or strain sensor, since its Bragg wavelength shifts with the temperature and strain variations, respectively, ΔT and Δε: (2)$${\Delta \lambda }_{B}=C_{T}\times \Delta T+C_{\varepsilon }\times \Delta \varepsilon$$
where $C_{T}$ is the temperature coefficient defined in a temperature range smaller than 100 °C (since the dependence of the Bragg wavelength on the temperature variations has to be described with a polynomial function in larger temperature ranges), and $C_{\varepsilon }$ is the strain coefficient [3]. For example, for an FBG written in a silica-based optical fiber and having a Bragg peak around 1550 nm, $C_{T}$ is about 10 pm/°C in the temperature range between RT and 100 °C [3], whereas the axial strain coefficient is around 1.2 pm/µε [45]. Concerning the sensitivity to the transverse load (or pressure), it is negligible for a classical FBG. 

As introduced in Section 0, gratings written in highly birefringent optical fibers are characterized by two Bragg peaks, which both shift towards the same direction under a temperature or axial strain change but shift in the opposite direction under a transverse mechanical load, making these FBGs very good pressure sensors with a sensitivity of hundreds of pm/(N/mm) [46].

## 3. Inscription Techniques

The periodical refractive index modulation, giving rise to the grating, originates from the silica matrix modifications induced by its exposure to laser light, which can generate defects or structural changes. To create this periodical structuration, several inscription techniques exist. 

The first method is known as “point by point” (briefly PbP) [47] and consists in focusing the laser beam through an objective on a point of the fiber core, to change its refractive index locally, then the fiber (or the laser beam) is translated for a certain distance, corresponding to the grating period (Λ), to write the following point. After repeating the process N times, a grating of length L=N×Λ is formed. 

The other techniques, by contrast, need an interferometric pattern, such as the one used to manufacture the first FBG by Hill [41] or the ones based on the “phase mask” (PM) [48], the Lloyd’s mirror [49], or the Talbot interferometer [50]. 

The phase mask technique is the most common today. It is based on a PM, which is a one-dimensional periodical structure, photolithographically etched on one of the surfaces of a slat of a material transparent to the laser light, such as silica glass. When the laser beam goes through the PM, the phase is spatially modulated and diffracted: the diffracted orders give rise to an interference pattern, with a period equal to half of the one of the PM (The amplitude of the PM structure is optimized in order to reduce the light transmitted in the zero order (less than 5%) and to divide the whole beam energy between the orders −1 and +1, or −2 and +2 [3]). This interference pattern creates the refractive index modulation in the fiber core of the optical fiber placed near the PM if the laser beam energy is high enough to create color centers and/or densification in the lightning area. In this case, the Bragg wavelength depends mainly on the PM period for fibers with the same composition.

The laser employed for the FBG inscription can be a continuous wave (CW) or pulsed laser in the domain of ns or fs. Whereas the wavelength of CW or ns-pulsed lasers has to be in the UV spectral range since one photon must be enough energetic to modify the silica matrix, this condition is not mandatory for fs-pulsed lasers. FBGs can be written with fs-lasers working in the spectral range from UV to IR: if the laser wavelength is in the IR domain, multi-photon processes can take place. 

Because of the fiber-coating shielding effects during the grating manufacturing especially with UV lasers, FBGs are often written in bare (uncoated) fibers: the coating is stripped before the grating inscription and reapplied just after. With the development of fs-laser-based techniques, today it is possible to inscribe FBGs also through the various types of coating [51]. 

Moreover, the inscription set-up can also be incorporated in the optical fiber drawing tower, so that the FBGs are written directly on the bare fiber before its coating deposition. Such FBGs are known as “draw tower gratings”, or DTGs [52].

## 4. FBG Classification

The FBG properties, such as their response in harsh environments characterized by extreme temperatures and/or radiations, depend on the fiber characteristics, inscription processes, and pre- or post-inscription treatments. Depending on the manufacturing choices, the gratings are generally classified into different types; each one is characterized by a different origin for the refractive index modulation, which gives rise to a different resistance to high temperatures and also to different radiation responses. 

As an example, by combining the PbP or PM technique and an fs-laser, both type I and type II gratings can be written in all fiber types. One FBG type or the other will be inscribed by increasing the laser power; indeed, for a fiber having a Ge-doped core, with an fs-laser at 800 nm and a PM, the thresholds of the pulse peak intensity to write type I or type II FBGs are $2\times 10^{13}\;W/{\mathrm{cm}}^{2}$ and $\sim 5\times 10^{13}\;W/{\mathrm{cm}}^{2}$, respectively [53]. These thresholds depend on the fiber composition, the possible fiber treatments undertaken to improve its photo-sensitivity (such as the H_2_ loading), the inscription technique, and the laser wavelength. 

Table 2 reports the most common grating types defined that have been defined, with the most important inscription conditions, the cause of the refractive index modification, and the maximum temperature each type can withstand, together with some of the most important references (for a more complete classification, see reference [54]).

## 5. Radiation Effects on Optical Fibers

As highlighted in Table 1, each harsh environment is characterized by different irradiation conditions:-Nature of radiation: X-rays, γ-rays, protons, electrons, neutrons;-Total ionizing dose (TID): quantity of energy deposited for the unit of mass of the material; it will be measured in Gy(SiO_2_) in all the manuscripts, except when specified differently;-Dose rate: quantity of energy deposited per unit of time, measured in Gy(SiO_2_)/s;-Irradiation temperature.

Radiation breaks bonds by ionization or knock-on processes. The main effect is the generation of point defects from regular or strained Si-O-Si bonds and from precursor centers. Their nature and concentration depend on the composition and also the manufacturing and drawing processes of the fiber; consequently, these parameters govern its radiation sensitivity. A list of the defects induced in silica-based OFs with their description and characteristics is reported in a review [61]. 

It has been demonstrated that ionization is the predominant effect even in the presence of non-ionizing radiation as neutrons, at least for fluences up to 10^16^ neutrons∙cm^−2^ (or n∙cm^−2^). Indeed, not only are the same defects generated independently of the radiation nature but also their concentration showed similar kinetics and levels at the same TID [62]. Nevertheless, some differences can be observed because of the dose enhancement induced by energetic proton recoils out of the H-containing coatings [63]. In addition to this effect, higher neutron fluences lead to significative structural changes, i.e., densification [16], causing new intrinsic defects or changes in the spectroscopic signatures of already known centers [64].

These phenomena at the microscopic level are at the origin of the degradation of the optical properties of the fibers and show up in three macroscopic effects, described hereafter.

### 5.1. Radiation-Induced Attenuation

The main effect induced by point defects is “radiation-induced attenuation”, or briefly RIA, which consists of an increase in the fiber attenuation due to the appearance of their associated absorption bands in the fiber transmission windows. RIA levels and kinetics depend on several parameters [15]:-Those of the harsh environment: dose, dose rate, temperature, and radiation nature, as already explained, but also the presence of gases that could diffuse inside the optical fiber;-The characteristics of the optical fiber itself: its core and cladding compositions, the manufacturing process of the preform, and fiber drawing conditions;-The test conditions: injected signal wavelength and power.

Clearly, fibers with different compositions present different radiation responses, and they are generally divided into three main classes [15], even if the radiation hardness of a fiber is only valid for specific environments and, sometimes, for specific wavelength ranges [65]:-The “radiation-hardened” OFs, having a pure-silica core (PSC) and fluorine-doped cladding or both core and cladding doped with F, since they show the lowest sensitivity under high-dose (MGy levels), steady-state irradiation among all the fiber types [61];-The “radiation-tolerant” OFs, such as Ge-doped ones, whose losses are low enough to be used for some applications, such as telecommunications, in harsh environments characterized by low TIDs (<10 kGy), i.e., space [14,66,67];-The “radiation-sensitive” OFs, which are mainly doped or co-doped with phosphorus or aluminum in their core and/or cladding and present high RIA levels, both in the visible and infrared spectral domains; they could be used for point or distributed radiation detection and dosimetry applications [68,69].

However, it is very important to test the radiation response of a fiber, before using it in a real environment. For example, it has been recently found an ultra-low losses PSC OF (whose losses at 1550 nm are of ~0.15 dB/km before irradiation) that is very radiation sensitive, with RIA levels higher than the ones induced in the P-doped fiber at 2 kGy TID (dose rate of 175 mGy/s) at RT: RIA reached ~2000 dB/km at 1550 nm [70]. 

### 5.2. Radiation-Induced Emission

The “radiation-induced emission”, or RIE, consists of two contributions: Cerenkov radiation [71] and radio-luminescence (RL) [15]. The latter is the emission of light from pre-existing or radiation-induced centers that will be excited by the radiation itself and that will emit a luminescence signal when coming back to the fundamental level. The Cerenkov light and also the RL, at least in most cases, affect the visible spectral range and should not interfere with the FBGs designed to operate in the IR spectral range.

### 5.3. Radiation-Induced Compaction

Concerning the “radiation-induced compaction”, or RIC, for the silica-based materials, it was observed that the density ρ increases with the dose D following a power law:(3)$$\frac{\Delta \rho }{\rho }\propto D^{k}$$
where k depends on the irradiation nature, being ~2/3 for γ-rays and ~1 for fast neutrons [72], before saturating around 3% at very high doses [16]. For the silica-based OFs, very few data are reported in the literature: linear compaction of ~0.3% (corresponding to a density increase of about 1%) was observed in pure silica samples irradiated with a fast neutron fluence of ~5 × 10^19^ n/cm² and a total γ-dose of about 4 GGy, at a temperature around 290 °C [23].

## 6. Radiation Effects on FBGs

As radiation degrades the fiber transmission through the RIA phenomenon, it can also degrade the FBG performance, as shown in Figure 2, through the two effects highlighted in Figure 3 [13].

The first one is the “radiation-induced Bragg wavelength shift”, hereafter indicated as RI-BWS. As shown in Equation (1), the Bragg wavelength depends on the effective refractive index at $\lambda_{B}$ and the grating period. The refractive index change is caused by the RIA and the RIC and is known as the “radiation-induced refractive index change” (RIRIC). The grating period modification, instead, can be reduced as a consequence of the compaction. As already mentioned, the latter effect is negligible in most cases; then, RI-BWS can be mainly associated with RIRIC. 

The second effect induced on the FBG is a variation of the grating reflectivity, caused by:

-Changes in the grating parameters, such as $n_{\mathrm{eff}}$, ${\Delta n\;}^{\mathrm{mod}}$ and Λ,-Degradation of the fiber transmission due to RIA.

Both can lead to a reduction in the signal-to-noise ratio and, in the worst cases, to an increase in the uncertainty associated with the measurement. At very high TID and/or neutron fluence, depending on the initial grating reflectivity, the peak can also be completely erased and the sensor stops working [24]. It is worth noticing that, if the periodical structuration is still present, the Bragg peak can be completely imperceptible within the noise in the transmission spectrum, but it should be detectable in the reflection one, despite its very small reflectivity. An example is reported in [24], where the reflectivity of an FBG (written in a Ge-doped fiber with an fs-laser at 800 nm and treated at 372 °C for 72 h) was reduced from 5% to 0.2% after a neutron fluence of 3 × 10^19^ n∙cm^−2^ (neutron energy of ~1 MeV) and a γ-dose of 1.5 GGy at 250 °C: the peak was still evident in reflection but not in transmission. 

Sometimes, instead, an increase in the reflectivity is observed, due to the different kinetics of radiation-induced defects in the laser-modified and unmodified (or weakly modified) areas of the grating (photoinduced refractive index peaks and valleys), as observed in [74]. Finally, when RIA is the cause of the reflectivity decrease because the fiber in which the FBG has been inscribed is characterized by a very high RIA level, the short fiber length containing the grating can be spliced to more radiation-hardened fiber pigtails to transport the signal to reduce the RIA impact [13]. 

In the next sections, we will focus on the RI-BWS. Since the Bragg peak translates into the value of the external parameter changes that the FBG is sensing, RI-BWS corresponds to an error in the measurement. For example, for an FBG having a thermal sensitivity coefficient $C_{T}$ of 10 pm/°C and a strain coefficient $C_{\varepsilon }$ of 1.2 pm/µε, a RI-BWS of about 10 pm will correspond to an error of about 1 °C or 8 µε in the temperature or strain measurements, respectively. Moreover, it has been demonstrated several times that the thermal sensitivity and strain coefficients are not significantly influenced by radiation [75].

As already mentioned, the grating is a periodical structuration of the refractive index induced by a laser in the fiber core. Since the laser modifies the silica matrix, e.g., generating defects and inducing densification, it is understandable that the radiation effects induced on the bright and dark fringes of the FBG can differ from the ones induced on the fiber itself, which did not undergo any laser treatment. Consequently, *the radiation response of an FBG cannot be directly linked to that of the fiber* and depends on several parameters [76]. Each FBG is particular as it has been written under specific conditions with particular characteristics; therefore, it will show a unique radiation response. For this reason, more than one hundred papers have been published concerning the radiation effects on different FBG types, and a review article has been published in 2013 by Gusarov and Hoeffgen [13]. Since then, new tests on new types of gratings have been performed and in the following, we present the main parameters that influence their radiation response and the associated basic mechanisms. It is worth noticing that, unless otherwise specified, the gratings were bare, meaning without a coating during the irradiation tests.

### 6.1. Optical Fiber Composition

As the radiation response of the OFs strongly depends on the compositions of their core and cladding, the fiber composition also influences the response of the gratings under irradiation. However, it is impossible to determine how, since the grating inscription process itself changes the fiber properties locally. This has been highlighted by Henschel et al., studying the BWS induced by γ-rays on type I-UV gratings inscribed in fibers with differently doped cores: germanium, phosphorous, and cerium [76]. Figure 4 shows a few of their results (extracted from [76]). It can be observed that the grating written in the radiation-sensitive (in terms of RIA) optical fiber co-doped with Al and P shows the smallest RI-BWS among all the reported gratings.

As a consequence, *writing a grating into a radiation-resistant fiber does not ensure a radiation-resistant FBG*. However, as Figure 4 suggests, it seems that the higher the GeO_2_ concentration in the fiber core, the higher the FBG radiation sensitivity [77,78]. This is highlighted in Figure 5, which compares the radiation responses of two 10 mm long gratings with the same reflectivity, written with a UV laser at 244 nm through a PM in two H_2_-loaded Ge-doped fibers (at 160 bars and RT, for 10 days): a standard Corning SMF28 (5 wt%) and a photosensitive (>15 wt%). The two gratings were also subjected to the same thermal treatment (at 60 °C for 24 h) to highlight only the Ge-content influence.

### 6.2. Pre-Inscription H_2_ Loading

Figure 5 also compares the BWS induced by X-rays on two gratings written with the UV laser in the same fiber, pre-hydrogenated or not before the FBG inscription. In this case, the two gratings were not identical because of the reduced photosensitivity of the unloaded highly Ge-doped fiber compared to the loaded one. Indeed, the FBG written in the H_2_-loaded fiber (with a laser power of ~90 mW) was only 10 mm long and had a reflectivity of 90% (transmission of −10 dB), whereas the grating inscribed in the unloaded fiber with a higher laser power (about 170 mW) was 20 mm long in order to reach a reflectivity of 50% (transmission of −3 dB). Conscious of these differences, it can be stated that *the H_2_ loading used to enhance the fiber sensitivity to UV light increases also the FBG radiation sensitivity*, independently of the fiber composition. With pre-hydrogenation, the RI-BWS saturates at higher levels and at higher doses [75,77]. The probable cause is the radiolytic rupture of the OH-bonds, which are present at higher concentrations in the gratings written in H_2_-loaded fiber [79].

This agrees with the following observation: the higher the H_2_ concentration in the fiber core at the time of the inscription, and consequently the OH concentration present in the grating, the more radiation-sensitive is the FBG. Indeed, Henschel et al. highlighted that the RI-BWS increases by increasing the pressure during the fiber H_2_ loading: since the fibers were loaded for the same duration and at the same temperature (one week at 50 °C), a higher pressure leads to a higher H_2_ concentration at saturation during the inscription [80].

Moreover, *the grating response does not change significantly, if the loading is performed with H_2_ or D_2_*, provided that the same concentration of molecules (H_2_ or D_2_) is reached at the moment of the grating inscription, as it was observed for regenerated gratings (no data are available for type I FBGs) [81]. The differences recorded in the RI-BWSs were within 10%.

Concerning the pre-hydrogenation, it must be pointed out that a grating written in a loaded fiber has to undergo thermal treatment to ensure the out-gassing of the remaining H_2_ molecules. For example, for the gratings, whose results under X-rays are reported in Figure 5, both were subjected to annealing at 60 °C for 24 h to outgas the remaining H_2_ molecules. This is an important point, since, as we will see in the next paragraph, the post-inscription thermal treatments also influence the FBG radiation response.

These conclusions about the H_2_-loading effect on the grating radiation sensitivity drawn from the type I gratings written with a UV-laser, for which the hydrogenation is necessary in most cases, can also be extended to other FBG types, as illustrated in Figure 6 for type I and type II FBGs manufactured with an fs-laser [82] under γ-rays. Indeed, it is worth mentioning that a different behavior will be highlighted under neutrons in Section 0.

### 6.3. Inscription Conditions

Beyond the H_2_-loading influence, Figure 6 highlights a clear difference in the radiation response of two gratings written with different laser powers. Indeed, among all the parameters, the radiation response of the grating strongly depends on its inscription conditions, which rule the grating type. Figure 7 reports the Bragg wavelength shift induced by X-rays on several FBG types written in the same Ge-doped optical fiber: the Corning SMF28e+ [74].

It should be pointed out that, for the type I FBGs and the seed grating of the regenerated FBG that were written with a UV laser, it was necessary to increase the fiber photosensitivity with pre-hydrogenation before their inscription, because of its low Ge-core concentration. As already highlighted in the previous section, the H_2_ loading increases the fiber radiation sensitivity but, as shown in Figure 7, even if the two gratings have the same length (10 mm) and are characterized by the same refractive index modulation amplitude ($\Delta n_{\mathrm{mod}}$ of about $1.2\times 10^{-4}$), the inscription conditions of type I gratings (laser wavelength and power) influence the radiation response, reaching different RI-BWS values by a factor of three. In particular, 150 pm and about 50 pm were observed, at 1 MGy TID, for the gratings inscribed with the 10 ns pulsed laser at 248 nm and the CW laser at 244 nm, respectively.

Moreover, in addition to the higher BWS values, the different types of FBGs present different kinetics for the RI-BWS. The type I UV grating is characterized by a RI-BWS increasing with the dose, without reaching a saturating level at 1 MGy TID. Both types II and III FBGs, instead, present a strong increase in the Bragg peak shift up to ~20 pm at about 50 kGy and then stabilize. Additionally, the type R grating shows a fast and strong RI-BWS increase at the irradiation start (40 pm for a TID of 50 kGy), but, contrary to types II and III, at higher TID the shift seems to increase linearly with the dose. In conclusion, at RT, type II and type III show the highest radiation tolerance, compared to type I or type R.

As for the RIA, the RI-BWS value reduces after irradiation: this phase is known as “recovery”. This effect is more or less important depending on many parameters, such as the inscription ones. In the case reported in Figure 7, the RI-BWS is reduced by about 15% for type I FBGs, 5% for type III, and more than 50% for type II. For type R, a recovery of around 7% has been observed (not reported here, since the TID reached in the experiment was 2 MGy) [83]. It is also worth noticing that the dose rate employed for the irradiation of the latter FBG is lower than the one used for the other samples; however, the RI-BWS should be larger at the higher dose rate, as highlighted in Section 0. Consequently, the change of this parameter in the comparison reported in Figure 7 does not invalidate our statements.

### 6.4. Bragg Wavelength

Among the inscription conditions, there is also the grating period, which rules the Bragg wavelength. Figure 8 reports the shift induced on three similar type I gratings, having the same amplitude of the refractive index modulation and submitted to the same treatments before and after inscription, but written at different Bragg wavelengths: 1300 nm, 1430 nm, and 1560 nm. By increasing the latter, the RI-BWS increases (from ~85 pm for 1300 nm to ~125 pm for 1560 nm at 1 MGy); however, once normalized by the initial Bragg wavelength of each grating (see inset of Figure 8), only a slight dependence can be observed, suggesting that the same variation of the refractive index occurs in all the gratings.

### 6.5. Post-Inscription Thermal Treatment

As for the pre-inscription treatments, e.g., the H_2_ loading, also the treatments performed on the FBG after its inscription influence its radiation response. The most common is a thermal treatment, used to stabilize the grating. Figure 9 gives some examples of the impact of such thermal treatments on the different FBG types.

The type I gratings in Figure 9a were written with an fs-laser at 800 nm in a Ge-doped fiber [84]. Indeed, since the grating inscription with such a laser does not require a photosensitive fiber, no H_2_ loading was performed. Consequently, the different effects due to a different thermal treatment performed after inscription and reported in Figure 9a cannot be associated with a phenomenon of hydrogen diffusion. For this FBG type, it is clear that by increasing the temperature of the pre-irradiation thermal treatment, the radiation sensitivity increases. Probably, this annealing recombines part of the defects generated during the grating manufacturing, creating more precursors available during the irradiation run.

Type R FBGs undergo high thermal treatment for the regeneration process. Consequently, thermal treatments at temperatures lower than the regeneration one do not change their radiation response (Figure 9c) [85].

The radiation sensitivity of type III gratings is also not influenced by thermal treatment, even if performed at very high temperatures, i.e., 750 °C, as shown in Figure 9d. This agrees with their stability at high temperatures [31].

Concerning the type II FBGs, in Figure 9b the pre-irradiation high-temperature treatment significantly improves its radiation resistance. Indeed, depending on the inscription conditions and on the quality of the set-up alignment, during the type II grating inscription, there can be regions (bright fringes) in which the laser intensity is above the threshold for type II and regions where it will be below this threshold. In these latter zones, only a type I modification will be generated. Consequently, the radiation response of such type II gratings will be influenced by its type I contribution. Thermal treatment at high temperature (~750 °C) can erase this latter contribution, based on point defects, leaving only the type II one, which results in a more radiation-resistant FBG [86].

### 6.6. Pre-Irradiation of the Grating

Contrary to the pre-thermal treatment, whose effect can be positive or negative according to the grating type, pre-irradiation seems to always have a positive effect, and indeed it reduces the radiation sensitivity of all the grating types [77], as shown in Figure 10. The pre-irradiation converts most of the precursors, making them unavailable for further irradiation. However, the amplitude of this effect depends on the grating type, the conditions of the pre-irradiation (mainly the TID), the conditions of the second irradiation, and above all, the conditions (temperature and duration) of the FBG storage between the two irradiation runs. 

### 6.7. Total Ionizing Dose

As shown in all the previous graphs, the RI-BWS depends on the dose. In most cases, the Bragg wavelength red-shifts with increasing dose: Sometimes the shift exhibits a fast increase at the irradiation start and then continues to grow with a smaller slope, and sometimes it quickly reaches a saturation. Such trends are similar to those observed for the RIA as a function of the dose, for which several models, both empirical and semi-empirical, i.e., the power law [90] and the fractal kinetics [91], have been developed to predict their growth kinetics (see reference [15] for an exhaustive list). Such models can also be applied to describe the RI-BWS versus dose curves, as presented in [78]. 

Although less frequent, cases have been reported in the literature where the Bragg wavelength shifts toward the shorter wavelengths with the increasing dose. The first example was observed by Gusarov et al. on the most classical type I FBGs, written with an ns-pulsed UV laser in an unloaded highly Ge-doped core that was not thermally treated, under γ-rays [92]. 

As illustrated in Figure 11, at the beginning of the γ-rays exposure, the Bragg peak shifts toward the red as expected, but then, after a TID of about 50 kGy, the direction of the shift is inverted and the peak shifts towards the blue. No detectable shift was observed on the gratings when kept at the same temperature as the irradiation, of about 34 °C, implying an instability of the grating. The authors suggested the presence of two different kinds of defects: one responsible for the refractive index increase and another for the index decrease. It is worth noticing that at the end of the irradiation run, the BWS continues decreasing.

Another example of a shift towards the shorter wavelengths has already been shown in Figure 9b on type II FBGs not thermally treated. The origin of this blue shift is very difficult to explain, but the hypothesis of a radiation-induced release of stress should also be considered, since the gratings were not thermally treated in both reported examples.

### 6.8. Dose-Rate

Concerning the dependence of the OF radiation response on the dose rate, it was observed that, in most cases, the higher the dose rate, the higher the RIA [93] at a given TID, since, by increasing the dose rate, the number of defects that could recombine during the irradiation is lowered. It has to be noted that this is not true for all the defects; some, such as the P-related P1 center absorbing around 1.5 µm, seem dose-rate independent, at least for TID up to 500 Gy [69].

A similar rule applies to the FBGs: *the higher the dose rate, the larger the RI-BWS* [94]. Of course, the amplitude of this effect depends on the FBG type, as highlighted in Figure 12. 

Type I and type R gratings generally follow this rule; however, the impact of this parameter on the RI-BWS depends on the fiber composition and the treatment the FBG goes through, as the regeneration temperature for the type R gratings. This is evident in Figure 12c for two types of R FBGs: the one written in a B/Ge co-doped fiber shows a higher dependence on the dose rate than the other one written in a Ge-doped fiber [81].

The type II FBGs, instead, are more radiation-resistant at RT and they are characterized by very low RI-BWSs. Consequently, the dose-rate dependence, if present, has not been highlighted yet, as reported in Figure 12b, where the RI-BWS varies between −5 pm and +5 pm, independently of dose and dose rate. The same conclusions can be deduced for the type III FBGs, in Figure 12d, even if a slight difference can be observed between the lowest investigated dose rate (1 Gy/s) and the highest one (40 Gy/s), in agreement with the general rule.

### 6.9. Irradiation Temperature

Combining temperature and radiation can lead to different effects [96]; for example, increasing the irradiation temperature could promote:-The thermal bleaching of the radiation-induced defects, reducing the RIA due to these color centers,-The conversion from unstable defects to more stable ones, giving rise to an RIA increase or a decrease depending on the investigated spectral range,-The defect generation rate, increasing the defect concentration and then the associated RIA at a given dose.

Obviously, the impact of temperature depends on time and consequently on dose and dose rate. For example, more time is needed with a lower dose rate than with a higher one,# to reach a certain TID; consequently, a center will have more time to recombine (thermal bleaching) or to be converted into another defect because of the temperature. Then, a lower concentration of this center will be observed at a lower dose rate. This illustrates how difficult the quantification of the impact of the temperature of irradiation is [96]. The grating response as a function of the irradiation temperature strongly depends on the grating type, as highlighted in Figure 13. For type I and type R (Figure 13a,c), the higher the temperature, the smaller the RI-BWS [80,81]. Concerning the type II FBGs, as for the dose rate, they show very low RI-BWSs at all the irradiation temperatures; for example in Figure 13b, the induced shift is between −10 pm and +10 pm at RT and at around 230 °C. The type III FBG, instead, showed a complex behavior. Despite its stability at high temperatures and its good radiation resistance, when the irradiation is performed at RT, it was observed that its radiation sensitivity increases by increasing the irradiation temperature [89], as reported in Figure 13d for a type III FBG written in the Corning SMF28e+. In this case, whereas at RT, the Bragg wavelength shifts towards the red and quickly saturates at 15 pm, after only 20 kGy TID (dose rate being of 40 Gy/s); at a high temperature of 250 °C, it blue-shifts down to −20 pm, without showing a saturating behavior, at least up to 1 MGy. Moreover, the gratings irradiated at high temperature lose their high temperature stability [89]. However, recently, a new result has been published, showing that the radiation resistance of type III FBGs improves by increasing the irradiation temperature, when the gratings are written with an optimized inscription set-up [87]. 

As the high temperature changes the radiation response of the gratings, a temperature lower than the RT value could also influence the induced BWS. Indeed, by changing the defect generation and recombination rate, at low temperatures, new centers could appear that were unstable at RT. Consequently, the RIA generally increases by decreasing the irradiation temperature, but obviously it depends on several parameters, among others, the fiber composition. For example, whereas the RIA at 1550 nm strongly increases at low temperatures for Ge-doped fibers (by a factor of 20 when the temperature decreases from RT to −80 °C at 10 kGy TID) [97], this effect is less dramatic for the F-doped fibers (with a factor of four, from RT to −80 °C at 100 kGy TID) [98]. Instead, the 1.55 µm RIA is stable within 15% from −80 °C and +120 °C for the P-doped OFs [99].

Figure 14 compares the BWSs induced on different FBGs when irradiated at RT and at a lower one, −120 °C. As for the high temperatures, the behavior changes with the grating types. The type I-UV here reported, an FBG written in an H_2_-loaded SMF28e+ with a UV CW laser (at 244 nm), shows a higher BWS [100], in agreement with the higher IR-RIA observed on the same Ge-doped fiber [97]. However, it should be pointed out that in [100] for another type I FBG written with an ns-pulsed UV laser in the same fiber, the BWS induced by X-rays at RT and at −120 °C showed similar kinetics and levels, leading to the conclusion that the response strongly depends on the inscription conditions. For type II and type III FBGs written in radiation-resistant fibers, instead, the BWS is very small at both the investigated temperatures, as illustrated in Figure 14b,c, showing that both FBG types are very radiation-resistant in this temperature range [100]. No data are available for the R-FBG behavior at temperatures below RT.

### 6.10. Nature of Radiation

When it comes to radiation, X-rays and γ-rays come to mind first. However, as reported in Table 1, harsh environments can be characterized also by the presence of other particles, such as protons, electrons, and neutrons. The first two are common in space, whereas neutrons are found in nuclear reactor cores or fusion-devoted facilities. Test facilities offering these types of beams are less accessible and only a few studies have been published about the FBG response under protons [39,101,102,103,104,105], electrons [104], or neutrons [19,22,23,24,27,28,106]. 

Even if particles can induce displacement damages, the response under radiation of a fiber depends on the defects created during the irradiation, and their main generation processes are associated with ionizing, at least for low to moderate fluences. Consequently, the RIA is mainly independent of the radiation nature [38] for most of the targeted environments (except for, e.g., nuclear core instrumentation). For example, the P1 is a center induced in the P-doped silica, absorbing around 1.5 µm: the growth kinetics of its absorption band is independent of the nature of irradiation, i.e., γ- or X-rays, protons, and neutrons, which makes it, together with other properties, a very good dosimeter of total ionizing dose in mixed environments [61,69,107].

Even if tests under γ- or X-rays can help us predict the RI-BWS, a comparison performed on the response of type I FBGs written with a UV laser in photosensitive fibers under protons or electrons and X-rays reported in [104] lets us conclude that the physics inside the gratings is more complex. It is worth noticing that the irradiations were carried out at the same dose rate and temperature up to the same TID, in order not to observe strange combined effects. For example, under protons of 63 MeV, at least up to 10 kGy (fluence of about 10^10^ p∙cm^−2^), the BWS induced on gratings written in Ge-doped or B/Ge co-doped fibers is slightly smaller than the one induced by X-rays under the same conditions. On the contrary, the response induced under electrons is strongly dependent on the fiber composition. Furthermore, as already mentioned, even if the radiation response of a P-doped fiber in the IR does not depend on the radiation nature, it is not the case for a grating written in a P/Ce co-doped fiber, where the X-rays induce a larger shift than electrons or protons.

Concerning the neutrons, whereas several campaigns have been carried out at low flux, up to a fluence of about 10^17^ n∙cm^−2^ [19,20,22,31], the research targets higher fluences for applications in nuclear reactor cores. The shift induced by a neutron fluence of 10^19^ n∙cm^−2^ (neutron energy of 1 MeV) at a temperature of about 250 °C with an accumulated gamma dose of 0.5 GGy was investigated on different gratings after irradiation [23,24]. Contrary to what was reported before, the gratings written in an H_2_-loaded Ge-doped fiber with an fs-laser were more radiation-resistant than the ones in unloaded fibers (all these FBGs were thermally treated at ~370 °C for 72 h). Indeed, the first ones showed a lower reflectivity reduction and also smaller BWS: this varies from −10 to −100 pm for the loaded FBGs, whereas it is about −130 pm for the unloaded ones. A type II FBG written in an unloaded Ge-doped fiber with an fs-laser and pre-treated at 750 °C for 2 h reached a BWS of −22 pm and −299 pm at the neutron fluence of 10^19^ n∙cm^−2^ and 3 × 10^19^ n∙cm^−2^, respectively. Moreover, it is worth noticing that writing such a grating in a pure silica core fiber pre-irradiated at 5 × 10^19^ n∙cm^−2^ and subjected to the same thermal treatment does not improve its radiation resistance, since the induced shift at the highest investigated fluence is still of about −200 pm.

Recently, FBGs were written in random air-line (RAL) fibers with an fs-laser (at 800 nm, with a pulse width of 80 fs) and tested under neutron fluence up to ~3 × 10^20^ n∙cm^−2^ (neutron energy of ~1 MeV), a flux of 6 × 10^13^ n∙cm^−2^∙s^−1^, and at a temperature of about 600 °C [26]. The Bragg wavelength showed a blue shift and a reflectivity reduction, both depending linearly on the time (or neutron fluence): their slope was about 0.096 nm/day and 0.125 dB/day, respectively. The BWS induced at the highest fluence was −4.47 nm. 

Such a large shift was also observed on a particular regenerated grating written in a Ge/F-codoped fiber. In [30], this is referred to as a chemical composition grating, or briefly, CCG, since the refractive index periodical structure is associated with the fluorine migration out of the bright fringes: the hydrogen present as OH groups (induced during the seed grating inscription in the H_2_-loaded fiber) will react with the fluorine, creating volatile HF molecules during the regeneration. Under neutron fluence up to ~10^19^ n∙cm^−2^ (neutron energy of 1 MeV) and flux of 7 × 10^12^ n∙cm^−2^∙s^−1^, with an associated γ-dose of ~2 GGy at 150 °C, this CCG showed a red shift with a saturating behavior up to ~14 nm, very different from the previous results that reported a blue shift.

### 6.11. Coating and Embedding

Most of the results here reported have been obtained on uncoated FBGs, in order to study only the radiation influence on the glass structure of the grating. However, for real applications, the gratings cannot be bare, since they will be more fragile. However, *the coating can influence the grating radiation sensitivity*. Gusarov et al. carried out a systematic study on the γ-rays’ effects on identical type I DTGs, written with a UV laser in a highly Ge-doped fiber during fiber drawing [108]. Their results are illustrated in Figure 15 and show that the BWS induced on the recoated FBGs is larger than in the bare sample. Indeed, whereas the shift in the latter is due to radiation effects on the glass, the difference between the coated and the bare FBGs has been attributed to the radiation effect on the coating: radiation makes the coating shrink or swell, inducing a change in the stress on the fiber and consequently a shift in the Bragg peak. From Figure 15, it is clear that the larger effects are observed for the ormocer and then the polyimide. The same authors also show that the shift induced on an acrylate-coated grating was between the ones recorded on the bare sample and the polyimide-coated one [108]. 

The same conclusions can be drawn from the study under proton irradiation by Curras et al. [103]. The authors explain the difference in the measured RI-BWSs with a two-stage process. Initially, the bonds of the coating polymeric chains are broken by radiation and induce a gas release, which will be trapped within the polymeric matrix, inducing the coating swelling. However, the gas release changes the polymeric structure, promoting transversal covalence bonds between the linear chains and consequently increasing the coating rigidity and inducing the coating shrinking.

However, in other studies under γ-rays [80], under X-rays [88], or under protons [105], the RI-BWS is smaller on coated gratings than on uncoated ones. In all cases, the coating does not shield the radiation (this could be the case for low-energy particles); moreover, it was demonstrated that the phase of UV curing (necessary for the bare FBG recoating with acrylate) does not influence its radiation response [88,109]. One of the remaining hypotheses is that the radiation continues the coating polymerization. Indeed, in [88], by performing at least three UV cycles on the recoated FBG, the authors show that such a recoated FBG will be characterized by the same RI-BWS as the bare grating. 

As with the coating, the embedding or packaging of the grating can also change its radiation sensitivity; an example of this is reported in [105]. Moreover, Lebel-Cormier et al. recently reported a radiation dosimeter based on polymer-embedded FBGs [110]: they showed that, by gluing the FBG inside a square prism in different polymer materials and reducing the error in the peak detection to 0.03 pm, the induced shift depends linearly on the dose, up to 20 Gy, with a slope of about ~0.06 pm/Gy, which depends on the material choice and not on the fiber composition. 

## 7. Radiation Effects on Exotic FBGs 

### 7.1. π-Phase Shifted Grating

A π-phase shifted grating is a grating with a phase step within the FBG length, which gives rise to a very narrow Lorentzian-shape band-pass peak (generally with a width of −3 dB of a few pm) in the band-stop spectral region of the FBG [111]. It is a very narrow filter, with a width of only a few pm; consequently, it can be used as a high-resolution sensor or spectral filter, to realize a distributed-feedback (or DFB) laser, an optical function for RF filtering.

During the inscription, a small birefringence can be induced on the fiber, inducing two peaks instead of only one. For this reason, these gratings are often written in polarization-maintaining fibers, which allow propagation of only one polarization and therefore excite only one peak. The radiation response of a π-phase shifted grating written in an H_2_-loaded Ge-doped core PANDA fiber with a UV laser and then subjected to thermal treatment at 120 °C for 8 h was studied under X-rays at RT up to 1 kGy TID, with a dose rate of ~30 mGy/s [112]. Both the main Bragg peak and the two peaks characterizing the π-phase shifted grating shift towards the red of the same quantity. For example, in this case, the BWS observed at the highest investigated TID was ~5 pm. 

It is worth noticing that the band-pass peak within the large Bragg notch depends on the phase-step size [111]; consequently, if this parameter does not change, the band-pass peak has to shift together with the main one. Nevertheless, it could also depend on the writing process used to incorporate the phase step in the grating. Moreover, at very high doses or neutron fluence, when a compaction phenomenon takes place [16], the phase-step size could change, leading to a different shift from the one induced on the main peak. 

### 7.2. Fiber Random Gratings

A fiber random grating, or FRG, is a grating whose refractive index modulation is not periodical but random [113]. An FRG is inscribed in the fiber core by the point-by-point technique, varying the distance between two consecutive spots randomly between 0 and a few µm, over a length of several cm, generally longer than an FBG [113]. Depending on the laser power density, type I or type II can be written.

Contrary to FBGs, the FRG gives rise to a very large reflection band, over 200 nm, caused by the superposition of the interference patterns of all the Mach–Zehnder interferometers constituted by each pair of two consecutive spots. A change in temperature or axial strain (but also in the surrounding refractive index) modifies the interferometer length and effective refractive indices of the core and cladding modes, causing a phase shift and, consequently, a spectral shift (*SS*) in the reflection spectrum [113], as shown in Figure 16. The spectral shift can be easily calculated by comparing two reflection spectra, acquired before and after a perturbation is applied, i.e., by performing a cross-correlation between the two. 

As for an FBG, the spectral shift depends linearly on temperature (in the 100 °C range) and strain:(4)$$\mathrm{SS}=C_{T}\times \Delta T+C_{\varepsilon }\times \Delta \varepsilon ,$$
with values of the same order of magnitude as those of a classic FBG for the temperature and strain coefficients, $C_{T}$ and $C_{\varepsilon }$, respectively [113]. However, the FRG can be employed as a multiparameter sensor, allowing the simultaneous measurement of the temperature and axial strain, through the “wavelength-division cross-correlation method”: the reflection spectrum has to be divided into N large subregions (N depending on the number of parameters to be measured, generally two or three), characterized by temperature and strain coefficients differing by up to 10%. The larger the subregions, the better the sensing resolution; however, the larger the spectral distance between the subregions, the larger the difference between the coefficients, and then the better the discrimination capability of the sensor [113]. This will allow writing a system of N equations, such as the one in Equation (4), with N unknown variables, one for each sensing parameter, ΔT, and Δε (more information about the analysis of simultaneous parameters is available in [115]). 

Type I and type II FRGs, written in a Ge-doped fiber, were tested under X-rays up to about 150 kGy, with a dose rate of 1 Gy/s, at RT [114]. Just as for an FBG, radiation induces an additional spectral shift, which depends on the fiber composition and inscription parameters. However, this shift is exactly the same for all the subregions. Therefore, for example with three subregions, it would be possible to discriminate two parameters even under irradiation, since one equation of the system will be sufficient to isolate the radiation-induced shift.

## 8. Radiation Effects on FBGs in Exotic Fibers

### 8.1. FBGs Inscribed in Highly Birefringent Photonic Crystal Fibers

Photonics crystal fibers, or briefly PCFs, are OFs characterized by a periodic transverse microstructure made up of air and glass (see review [116]). This structure allows the manufacture of highly birefringent fibers, and a grating written in it presents two Bragg peaks when investigated with an unpolarized light [117]. Indeed, the two orthogonally polarized modes propagate with two different phase velocities and consequently are characterized by different effective refractive indices, which depend on the mean effective index ($n_{eff}$) and on the phase modal birefringence *B*. Therefore, the spectral distance between the two peaks is defined as [117]: (5)$$\Delta \lambda_{B}=\lambda_{B2}-\lambda_{B1}=2\times B\times \Lambda .$$

Whereas the two peaks shift towards the same direction, keeping $\Delta \lambda_{B}$ constant, when the grating is subjected to a temperature or a longitudinal strain, under pressure or a transverse mechanical load, they move in opposite directions, reducing their spectral distance, as illustrated in Figure 17.

This characteristic makes this FBG a good pressure and transverse strain sensor for structural health monitoring [46]. It is worth noticing that such gratings are characterized by classical values for temperature and strain coefficients, whereas the transverse sensitivity varies between 150 pm/(N/mm) and 550 pm/(N/mm), depending on the fiber orientation during the compression test [46] (for more information, see review [118]). 

The radiation response of such a grating was investigated under X-rays [73]. The FBG was written in a highly birefringent PCF known as ‘butterfly’, having a core in Ge-doped silica to facilitate the grating inscription, with an fs laser at 267 nm, and underwent a 16 h thermal treatment at 80 °C. The grating presented two peaks with a spectral distance of ~0.74 nm, which corresponds to a phase modal birefringence B of ~7 × 10^−4^. This FBG was irradiated at RT up to 1.5 MGy TID, with a dose rate of 24 Gy/s. As observed for type I gratings, both Bragg peaks shift towards longer wavelengths during the irradiation, as shown in Figure 18, quickly reaching the saturation value at a TID of about 15 kGy. The spectral distance, instead, remains unaffected by the radiation, within 10 pm; such a small change corresponds to a variation of the phase modal birefringence B of ~10^−5^. Moreover, if this grating is used as a transverse mechanical load sensor, the radiation-induced error in the load measurement is less than 0.1 N/mm. In conclusion, *this transverse mechanical load sensor is intrinsically radiation-resistant up to MGy dose levels*.

### 8.2. FBGs Inscribed in Multicore Fibers

Multicore fibers, or MCFs, consist of fibers where multiple separate cores have been incorporated in their claddings. They were proposed and manufactured, and their optical properties were characterized, for the first time in 1979 [119]: seven preforms were inserted in a jacket tube, that was collapsed to create an unique preform before the fiber drawing process. Their main aim was to increase the fiber transmission capability for telecommunication. In this case, each core should be an individual waveguide, so the crosstalk was a phenomenon to avoid. The crosstalk depends on the distance between the fiber cores and the transmission distance and it consists in coupling a signal launched into one core with the neighboring ones. Nowadays, the crosstalk can be suppressed or enhanced (see review [120]). 

MCFs can have homogeneous or heterogeneous cores: in the first case, the refractive index profile of each core is the same, whereas in the second it is not and then the cores can present different types of doping. 

Such fibers have also attracted more and more interest for sensing applications, both as distributed sensors (see review [121]) and as point sensors based on FBGs (see review [122]), i.e., to discriminate strain and temperature or for shape sensing. Indeed, during fiber bending, whereas the central core (as the one in a classical SMF) is on a strain neutral axis and then insensitive, the cores on the outer and inner sides will be, respectively, stretched and compressed, so they will undergo a strain change. Consequently, monitoring the strain variation in at least three cores, for example with three FBGs, will give rise to a three-axis shape sensor [123].

Writing FBGs in these types of fiber is more complicated because of their geometry, as with the PCFs. With a UV laser and the phase mask techniques, gratings will be inscribed in all cores at the same time (if they are sufficiently photo-sensitive); instead, to write an FBG in only one core, an fs-laser should be used with the point-by-point technique [122].

Under radiation, the Bragg peaks of identical gratings written in all the cores of a homogeneous MCF have to shift by the same amount; consequently, *the shape sensors based on FBGs inscribed on homogenous MCFs are intrinsically radiation-resistant.* This was confirmed by Barrera et al. investigating the response under γ-rays of two shape sensors based on Bragg gratings written with an UV laser in two 7 Ge-doped cores MCFs, one H_2_-loaded, in order to increase the radiation sensitivityof each FBG, and one unloaded [124]. 

### 8.3. FBGs Inscribed in Polymer Optical Fibers

Polymer Optical Fibers (POFs), also known as plastic fibers, have several advantages compared to silica fibers, such as lower cost and higher flexibility, despite their higher optical losses [125]. Consequently, the Bragg gratings (or the ‘quasi-single mode’ Bragg gratings, since most of these fibers are multimode [126]) written in such fibers, briefly indicated as POFBGs, are interesting for several applications, thanks to their higher strain, temperature and humidity sensitivity (see review [127]). Even if more studies have been dedicated to the radiation-effects on the POF properties, also the radiation response of the FBGs inscribed in these fibers has been investigated by some researchers. 

As a first example, we report here a study about the response of a FBG written with an fs-laser in the CYTOP POF, which presents lower intrinsic losses at 1550 nm than the other types of POFs [38]. In this work, Broadways et al. observed, despite a peak shape change with the appearing of two sub-peaks, a small amplitude reduction of only 3 dB and a blue shift under γ-rays up to ~40 kGy TID, as illustrated in Figure 19.

The grating sensitivity was −26.2 pm/kGy, with a maximum BWS of about −1 nm at the highest investigated dose. Such a shift corresponds to a 70% increase in relative humidity or to a variation of 700 µε or 55 °C if the grating is used as a humidity, strain, or temperature sensor, respectively. However, the authors suggest its use as a possible dosimeter, whose resolution was about 40 Gy with their acquisition system. Figure 19 highlights a clear change in the peak amplitude of about 33 kGy: this was explained by a degradation of the splice between the CYTOP sample and the silica-based transport fibers used to connect the FBG to the acquisition system. Indeed, splicing still remains one of the most difficult tasks when working with POFs. 

It is worth noticing that sometimes a red shift for the POFBGs is observed. Indeed, in a work dealing with the response of an FBG written with a CW UV laser (at 325 nm) in a PMMA fiber under neutron fluence with energy from 2 MeV to 10 MeV [128], the Bragg peak shifts towards the red and reaches 7 pm at the highest investigated TID of about 720 Gy.

## 9. Radiation-Resistant FBGs for Applications in Harsh Environments

All the studies published until now, and here summarized, on the radiation response of the FBGs, allow us to conclude which grating type will be the best choice for a given application in a given radiation-rich environment. It is very important to keep in mind that writing a grating in a radiation-resistant fiber does not assure its radiation resistance. However, using radiation-resistant or tolerant fibers will reduce the peak reflectivity decreases associated with the RIA phenomena.

As reported in Table 1, each environment is characterized by different doses, dose rates, and temperatures, parameters that will influence the various FBGs differently.

The type I gratings, especially the ones written with a UV laser, are the easier ones to manufacture but are also the ones with larger RI-BWS. By decreasing dose and dose rate, this shift reduces, and this might suggest them as an acceptable option for low doses, e.g., for space applications. However, the very low temperatures that can be reached in such environments will worsen the sensor response, inducing bigger errors in the measurements. Even in this case, they are not an adequate choice. A solution to improve their performance could be pre-irradiation, but, in this case, their response will depend on the conditions of such a treatment and of the storage between pre-irradiation and real use: too many parameters to control. In conclusion, type I FBGs are not suitable for radiation-rich environments.

The regenerated gratings have been considered for nuclear reactor applications, which means high dose, dose rate, and temperature, from RT to 800 °C, with a high neutron fluence to complicate the situation. Even if very small shifts have been observed at high temperatures, it is not always the case at temperatures as low as RT: in all the graphs here reported, a RI-BWS of at least 40 pm has been observed on different R-FBGs, which corresponds to an error in the temperature measurement of about 4 °C. Almost no tests have been performed under a high neutron fluence. The only one reported by Fernandez et al. showed a shift of about 10 nm at a neutron fluence of ~10^19^ n∙cm^−2^ (neutron energy of ~1 MeV) with an associated γ-dose of ~2 GGy at 150 °C [30]. Consequently, until new tests are carried out, the radiation resistance of such gratings is not guaranteed.

Two grating types are left.

Type II gratings, written with an fs-laser, showed good radiation resistance, but this can be strongly dependent on the inscription conditions and set-up, especially when they are inscribed through a phase mask. With a PM, indeed, all the bright fringes are not illuminated with the same laser intensity, supposing that the laser beam is Gaussian. Therefore, as explained in Section 0, both type I and type II components could be generated. In order to obtain only a type II FBG, after its inscription in a radiation-resistant fiber with the optimized power conditions necessary to obtain a type II [53], thermal treatment should be performed to erase the type I component and improve the radiation resistance of the final FBG [86]. The good performance of such a grating was tested up to 3 MGy TID under X-rays or γ-rays, with temperatures from −100 °C to +350 °C, under protons for space applications, and even in a nuclear reactor core with a fast neutron fluence of 3 × 10^19^ n/cm² and a total γ-dose of about 4 GGy, at a temperature of about 290 °C (in this case, a RI-BWS of 40 pm was measured after irradiation and corresponds to a temperature error of only ~4 °C [23]). 

Finally, the type III gratings seem promising but still more investigations are needed. RI-BWSs of the order of magnitude of 10 pm have been observed on Ge-doped fiber (slightly lower on an F-doped fiber) at 1 MGy TID. Their response does not seem to change drastically with the dose rate or at low temperatures. However, what still remains to understand is their behavior at high temperatures. Indeed, a blue shift, almost linear with the dose, was observed during irradiation at a high temperature, about 200 °C, [89] whereas in another study, the grating showed a good radiation resistance [87]. The two investigated type III gratings were manufactured by different laboratories, and until now it is not clear how they differed. So, we cannot conclude yet which is a good process to obtain radiation-resistant type III FBGs.

## 10. Conclusions

Fiber Bragg gratings are point sensors suitable for monitoring temperature and/or strain. Various types exist with different inscription processes and different thermal and radiation resistances. Under radiation, indeed, the performance of such sensors can be degraded and an error in measurements can be induced. The magnitude of this error will depend on the conditions of the radiation-rich environment and on the grating itself. The purpose of this article was not to list all the published works about gratings under radiation—they are too many–but to provide a guide on how the radiation response of different grating types depends on the parameters of the environment, the FBG manufacturing, or the treatments performed before or after the inscription. It is important to highlight first that writing a grating into a radiation-resistant fiber does not ensure a radiation-resistant FBG, and second that the radiation resistance of an FBG strongly depends on its application. For example, a grating whose radiation response could be acceptable for a space application, as a type I, could not resist the environment of the nuclear reactor core. Consequently, it is not possible to determine a classification of FBGs based on their radiation response. However, among all the FBG types, the type II and the type III gratings seem to be the best option for applications having to operate in a wide temperature range and at high radiation doses. Furthermore, it is worth mentioning that, if the fiber is not radiation-resistant, the signal will not be transmitted from/to the instrumentation, because of the RIA. A solution, in this case, can be to splice a small piece of this fiber containing the FBG with a radiation-hardened one. 

To conclude, despite the fact that radiation alters the performance of FBGs, thanks to the various hardening studies led by the scientific community, radiation-hardened FBG-based sensors appear today as a very promising solution for monitoring environmental parameters in the most challenging radiation environments. Thanks to a better understanding of the basic mechanisms related to FBG photoinscription and radiation effects, we could imagine that even more radiation-tolerant FBGs could be designed in the future, also able to survive combined high temperature and high radiation dose constraints.

## Figures and Tables

**Figure 1 sensors-22-08175-f001:**
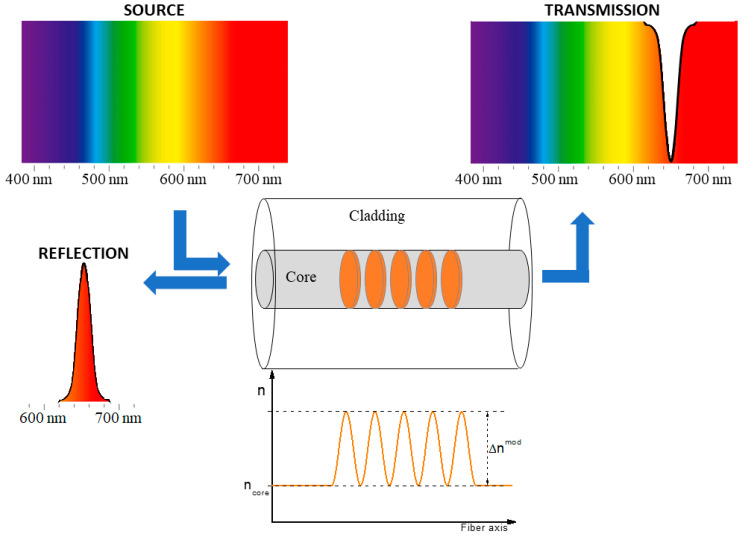
Transmission and reflection spectra of a fiber Bragg grating, with the periodical structure of the refractive index in the fiber core.

**Figure 2 sensors-22-08175-f002:**
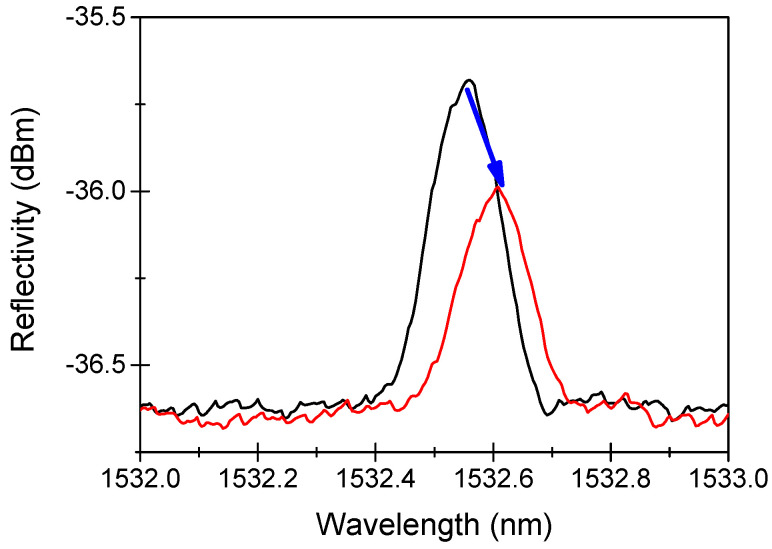
An example of the radiation-induced effects on a Bragg peak reflection spectrum, before (black line) and after (red line) X-ray irradiation at 1.5 MGy TID at RT. Data extracted from [73].

**Figure 3 sensors-22-08175-f003:**
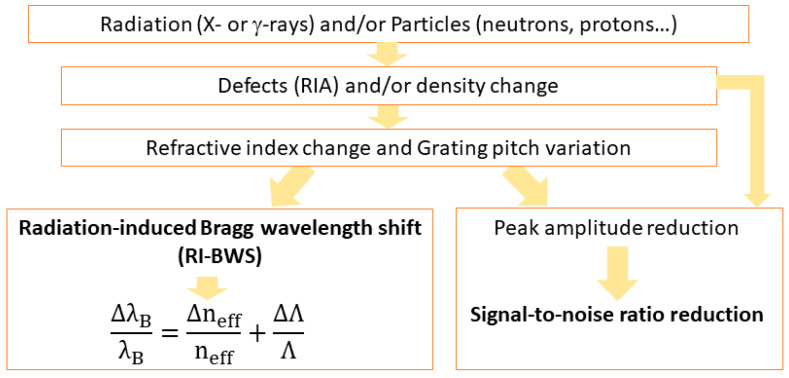
Origins of the radiation effects on the FBG peak: radiation-induced Bragg wavelength shift (RI-BWS) and signal-to-noise reduction.

**Figure 4 sensors-22-08175-f004:**
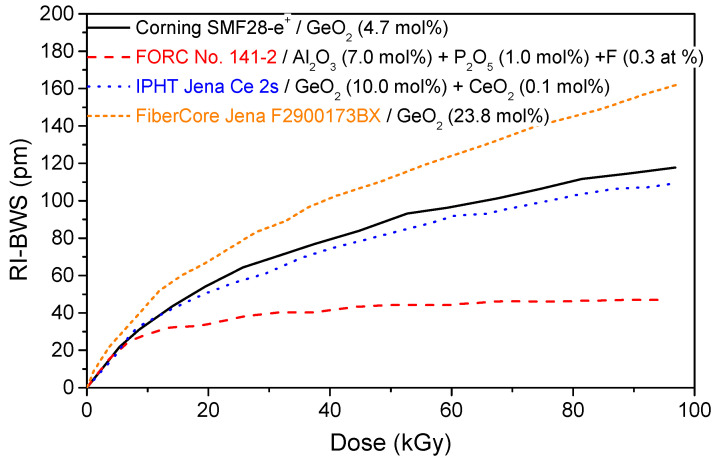
BWS induced by γ-rays on type I-UV gratings, written by different manufacturers on uncoated fibers with different core compositions that were recoated after inscription, the dose rate being 0.9 Gy/s up to 100 kGy at RT. Data extracted from [76].

**Figure 5 sensors-22-08175-f005:**
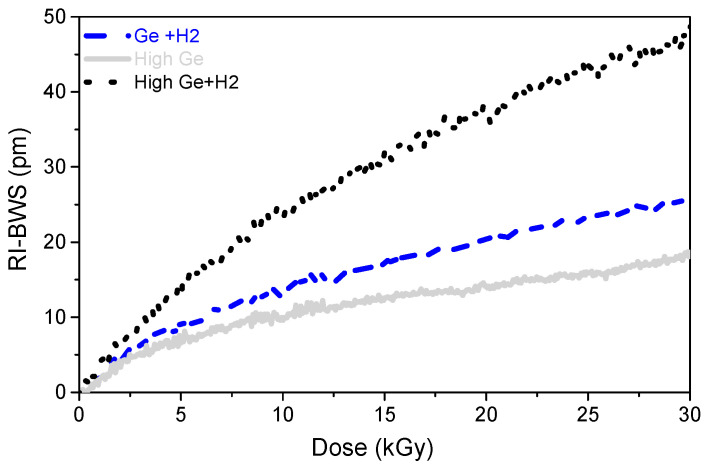
BWS induced by X-rays at RT with a dose rate of 5 Gy/s, up to 30 kGy, on three type I gratings written with a CW UV laser (at 244 nm) into two different fibers: a standard fiber (the Corning SMF28, [Ge] = 5 wt%) and a photosensitive one ([Ge] > 15 wt%). Ten-millimeter-long FBGs were written in both fibers after a pre-inscription H_2_ loading. A 20 mm long FBG was also written in the unloaded photosensitive fiber with higher laser power (170 mW against ~90 mW). Data extracted from [78].

**Figure 6 sensors-22-08175-f006:**
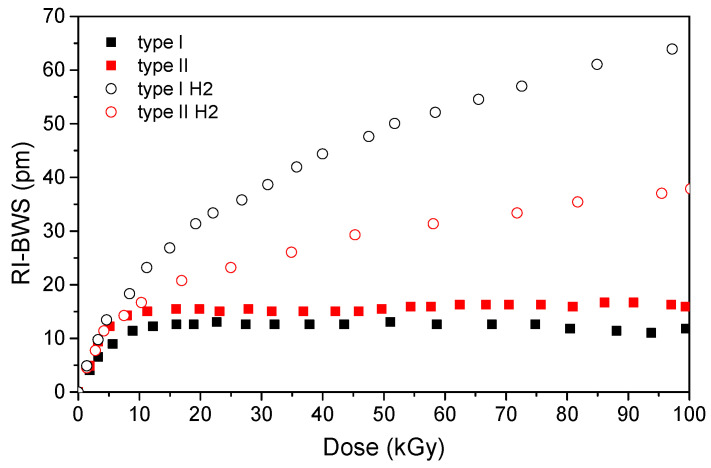
BWS induced by γ-rays at RT with a dose rate of 0.94 Gy/s, up to 100 kGy TID (irradiation temperature between 25 °C and 65 °C), on two type I and two type II gratings written with an fs-laser (at 800 nm, a pulse width of 120 fs, and repetition rate of 100 Hz) into the acrylate coated Ge-doped fiber Corning SMF28e. For each grating type, one FBG was inscribed into an H_2_-loaded fiber and the other in an unloaded one. All the gratings were thermally treated for 4 days at 100 °C. To write type I FBG, the laser power was reduced by increasing the distance between the PM and the fiber. Data extracted from [82].

**Figure 7 sensors-22-08175-f007:**
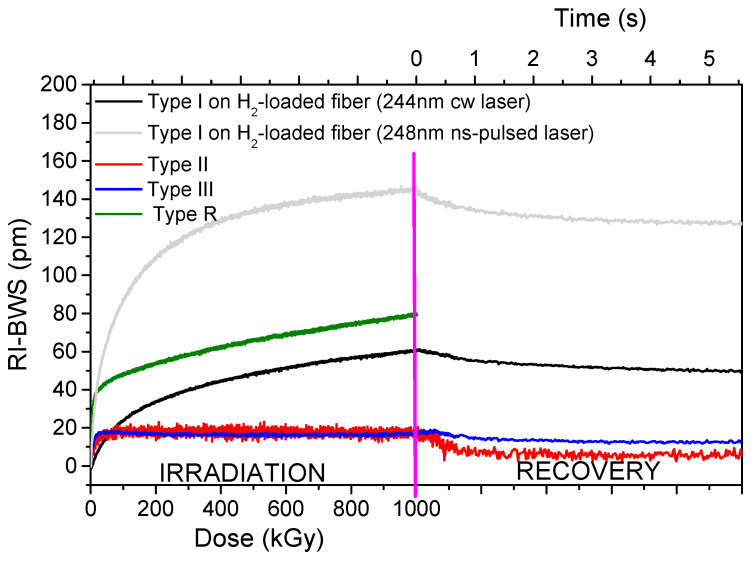
RI-BWS recorded up to 1 MGy at RT on different FBG types written in the same Corning SMF28e+ fiber: two type I FBGs, one written with a CW UV laser (at 244 nm, power density of 4.15 W/cm^2^) and the other with a 10 ns-pulsed UV laser (at 248 nm, pulse energy of 380 mJ/cm²), in a pre-H_2_-loaded sample; a type II FBG, written through a PM with an fs-IR laser (at 800 nm, a pulse width of 150 fs, and pulse energy of 560 µJ); a type III FBG, written PbP with an fs-IR laser (at 800 nm, a pulse width < 120 fs, and pulse energy of 230 nJ); an R-FBG, manufactured from a type I grating written with a 20 ns-pulsed UV laser (at 248 nm and pulse energy of 9 mJ) and regenerated at 750 °C. The vertical pink line indicates the irradiation end. The dose rate was always 50 Gy/s for all the gratings except for the regenerated one at 10 Gy/s. Data extracted from [74,83].

**Figure 8 sensors-22-08175-f008:**
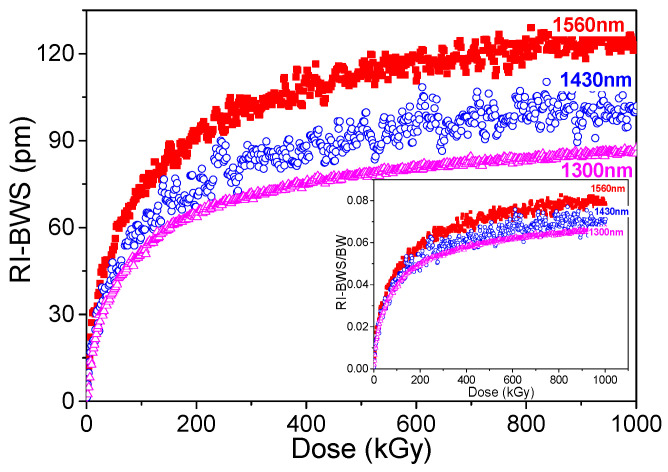
BWS induced by X-rays at RT up to 1 MGy (dose rate of 50 Gy/s) on three similar gratings (having the same $\Delta n^{\mathrm{mod}}$ of about $2\times 10^{-4}$ ) written with a CW UV laser (at 244 nm) in H_2_-loaded Corning SMF28 at different Bragg wavelengths. In the inset, RI-BWS normalized with respect to the initial Bragg wavelength. Data extracted from [49].

**Figure 9 sensors-22-08175-f009:**
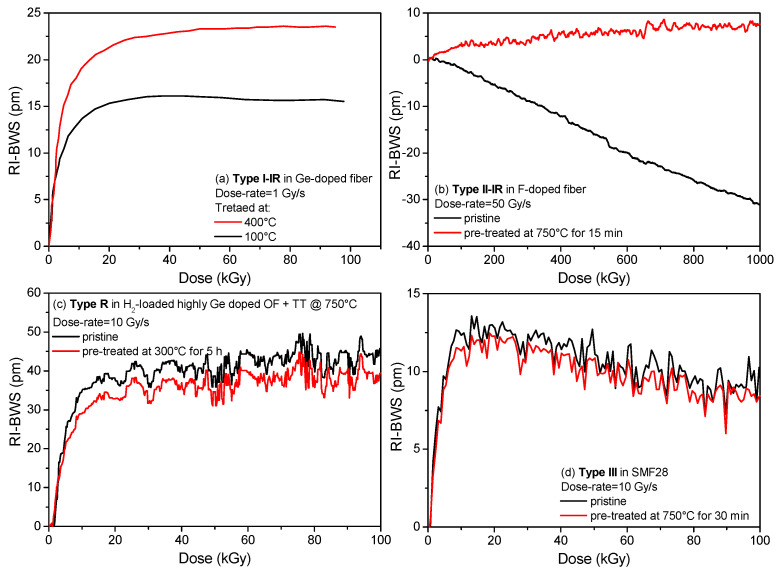
Effect of a pre-thermal treatment on the RI-BWS induced on different FBG types: (**a**) a type I FBG written in a Ge-doped fiber (the Corning SMF28e) with an IR fs-laser (at 800 nm, pulse energy of 1200 µJ); (**b**) a type II FBG in an F-doped fiber with an fs-IR laser (at 800 nm, a pulse width of 150 fs, and pulse energy of ~500 µJ); (**c**) type R FBGs written in H_2_-loaded highly Ge-doped fiber and regenerated at 750 °C; (**d**) a type III in the SMF28 fiber written PbP with an fs-IR laser (at 515 nm, with a pulse width < 290 fs). The gratings reported in (**a**) were investigated under γ-rays, whereas all the others were investigated under X-rays. Data extracted from [84,85,86,87].

**Figure 10 sensors-22-08175-f010:**
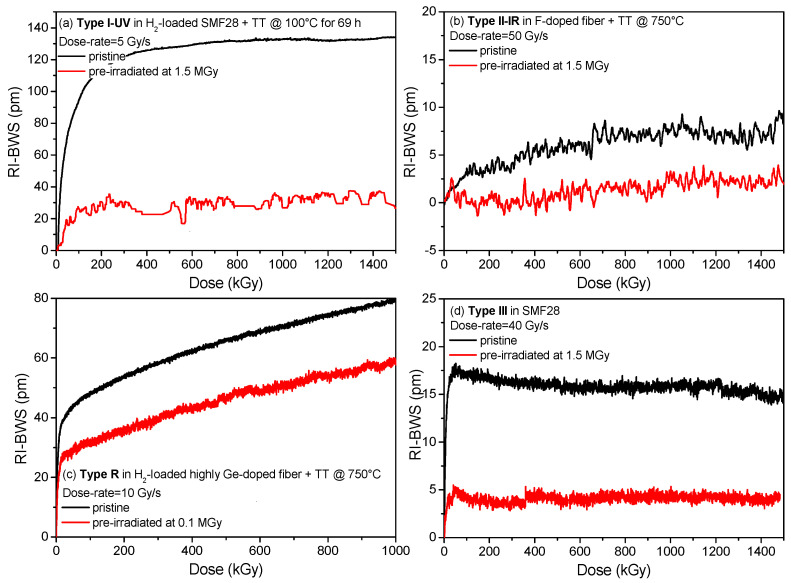
Effect of a pre-irradiation on the BWS induced by X-rays on different FBG types: (**a**) a type I FBG written in an H_2_-loaded SMF28 fiber with a UV CW laser (at 244 nm, power of 120 mW); (**b**) a type II FBG in an F-doped fiber with an fs-IR laser (at 800 nm, a pulse width of 150 fs, and pulse energy of ~500 µJ); (**c**) type R FBGs written in H_2_-loaded highly Ge-doped fiber and regenerated at 750 °C; (**d**) a type III in the SMF28 fiber written PbP with an fs-IR laser (at 800 nm, a pulse width < 120 fs, and pulse energy of 230 nJ). Data extracted from [83,86,88,89].

**Figure 11 sensors-22-08175-f011:**
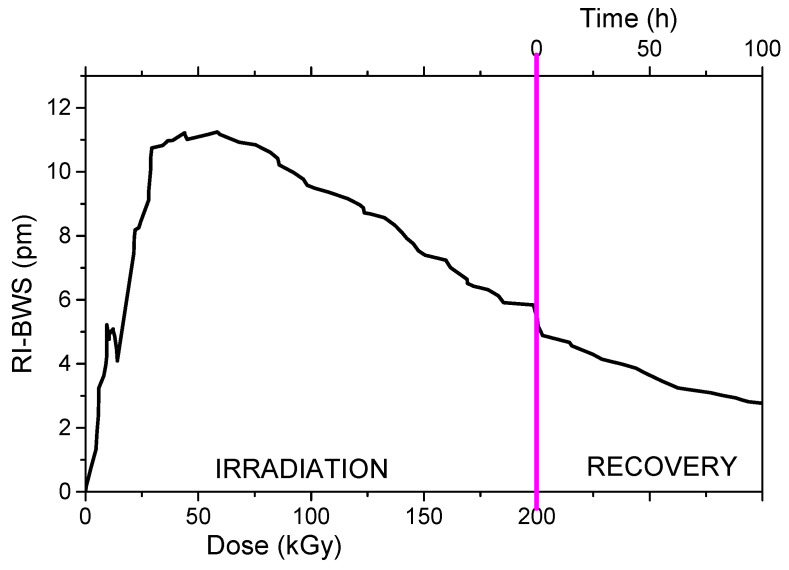
γ-rays-induced Bragg wavelength shift on type I FBGs, inscribed with an ns-pulsed UV laser (at 248 nm, a pulse width of 15 ns, and pulse energy of 8 mJ) in an unloaded highly Ge-doped core (~8 mol% of GeO_2_). The vertical pink line indicates the irradiation end. The accumulated dose reached in the only first irradiation run, here reported, was around 200 kGy, the dose rate being of about 0.28 Gy/s and a temperature of about 34 °C. Data extracted from [92].

**Figure 12 sensors-22-08175-f012:**
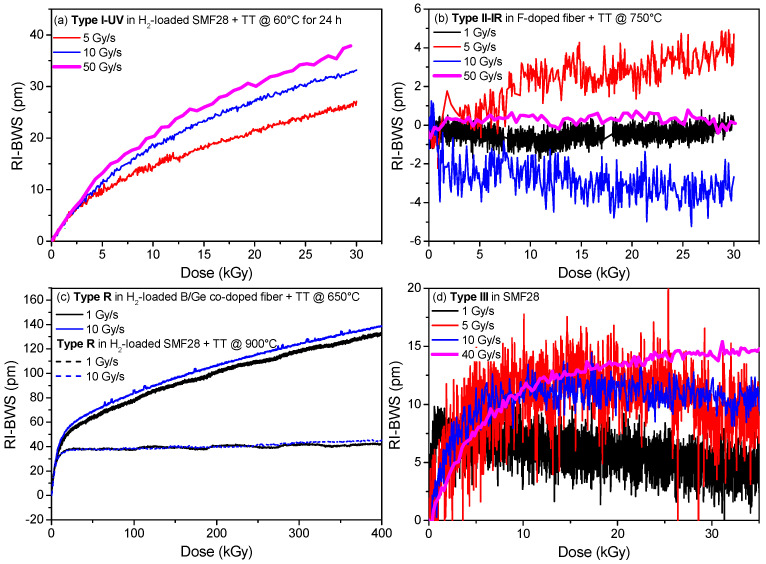
BWS induced by X-rays as a function of the dose, for different dose rates and different FBG types: (**a**) a type I FBG written in an H_2_-loaded SMF28 fiber with a UV CW laser (at 244 nm, power of 120 mW) and treated at 60 °C for 24 h; (**b**) a type II FBG in an F-doped fiber with an fs-IR laser (at 800 nm, a pulse width of 150 fs, and power density of $3\times 10^{13}\;W/{\mathrm{cm}}^{2}$) treated at 750 °C for 15 min; (**c**) type R FBGs in H_2_-loaded B/Ge codoped fiber (continuous lines) or SMF28 fiber (dashed lines) regenerated at 650 °C or 900 °C, respectively; (**d**) a type III in the SMF28 fiber written PbP with an fs-IR laser (at 800 nm, a pulse width < 120 fs, and pulse energy of 230 nJ). Data extracted from [78,81,89,95].

**Figure 13 sensors-22-08175-f013:**
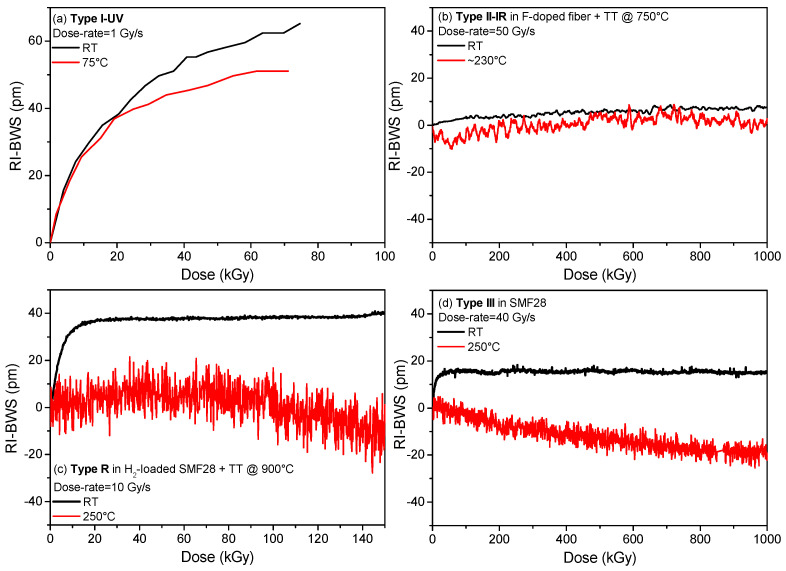
RI- BWS as a function of the dose, at two different temperatures, for different FBG types: (**a**) a type I FBG written with a UV laser (at 248 nm) in an H_2_-loaded Ge-doped fiber (SMF28e) treated at 240 °C for 3 min and at 100 °C for 3 days, with a dose rate of about 1 Gy/s; (**b**) a type II FBG in an F-doped fiber with an fs-IR laser (centered at 800 nm, with a pulse width of 50 fs) treated at 750 °C for 15 min, with a dose rate of 50 Gy/s; (**c**) type R FBGs in SMF28 fiber regenerated at 900 °C, with a dose rate of 10 Gy/s; (**d**) a type III in the SMF28 fiber written PbP with an fs-IR laser (at 800 nm, a pulse width < 120 fs, and pulse energy of 230 nJ), with a dose rate of 40 Gy/s. The gratings reported in (**a**) were investigated under γ-rays, whereas all the others were obtained under X-rays. Data extracted from [80,81,86,89].

**Figure 14 sensors-22-08175-f014:**
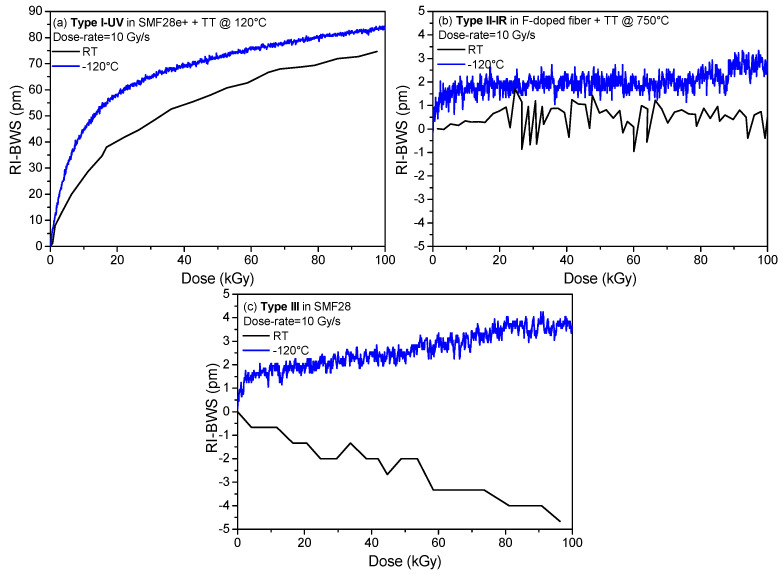
BWS induced by X-rays as a function of the dose at two different temperatures, for different FBG types: (**a**) a type I FBG written with a UV laser (at 244 nm) in an H_2_-loaded Ge-doped fiber (SMF28e) treated at 120 °C for 8 h, with a dose rate of about 1 Gy/s; (**b**) a type II FBG in an F-doped fiber with an fs-IR laser (at 800 nm, with a pulse width of 50 fs) treated at 750 °C for 15 min, with a dose rate of 10 Gy/s; (**c**) a type III in the same F-doped OF written PbP with an fs-IR laser (centered at 800 nm, with a pulse width 120 fs), with a dose rate of 10 Gy/s. Data extracted from [100].

**Figure 15 sensors-22-08175-f015:**
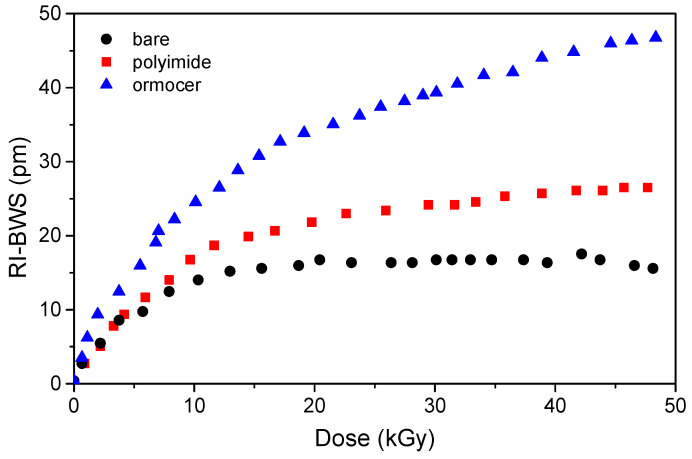
γ-rays induced BWS as a function of the dose, for identical type I FBGs written with a UV laser (at 248 nm, a pulse width of 15 ns, and power energy of 400 mJ/cm²) in a highly Ge-doped fiber and recoated after inscription with polyimide (which was thermally cured) or ormocer, with a dose rate of ~0.11 Gy/s, at 35 °C. A bare grating was also reported for the sake of comparison. Data extracted from [108].

**Figure 16 sensors-22-08175-f016:**
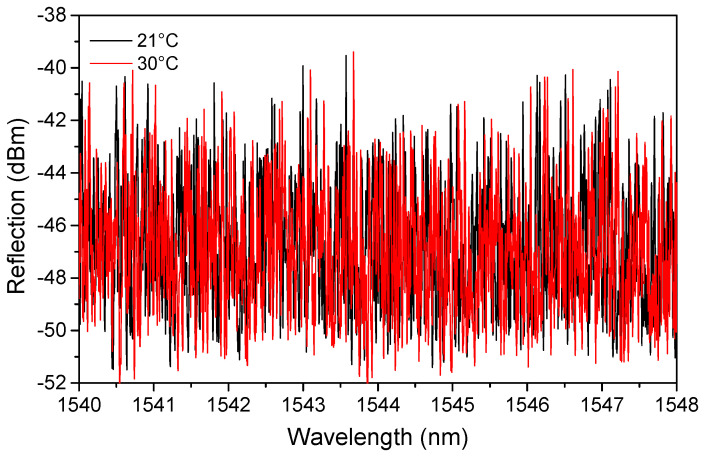
A part of reflection spectra of a type II 10 cm long FRG, inscribed in a Ge-doped fiber (the Corning SMF28e) with an fs-laser (at 800 nm), acquired at two temperatures, in order to highlight the changes induced by a temperature variation of about 10 °C. Data extracted from [114].

**Figure 17 sensors-22-08175-f017:**
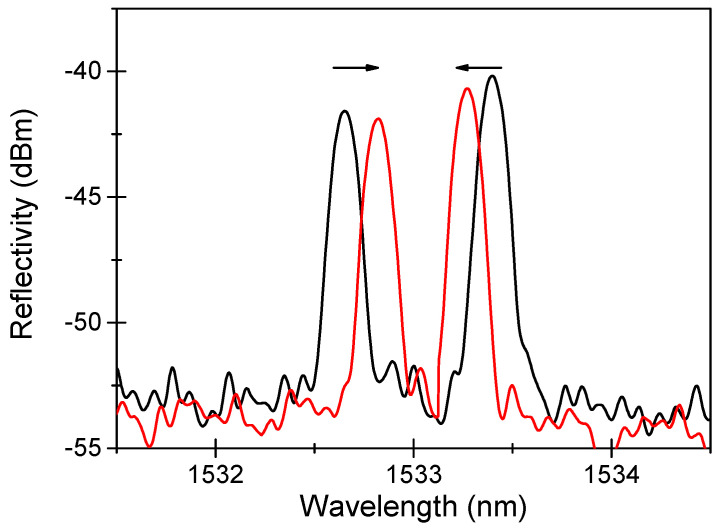
Reflection spectra of an FBG written in a highly birefringent PCF, unstressed (black curve), and when a transverse load is applied (red curve). The arrows highlight the direction of the shift in the peaks.

**Figure 18 sensors-22-08175-f018:**
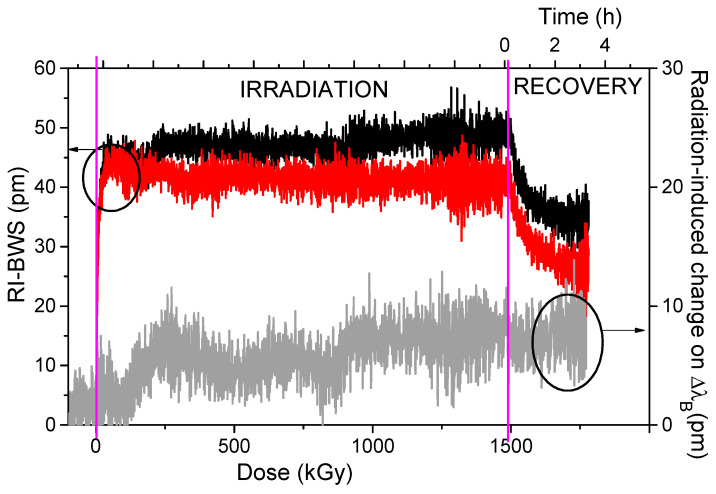
On the left axis, BWS induced by X-rays on the two peaks (peak 1—black points; peak 2—red points) of a grating written with an fs laser (at 267 nm and a pulse width of 120 fs) inside a butterfly PCF with a Ge-doped core, at RT up to 1.5 MGy TID, with a dose rate of 24 Gy/s. On the right axis, changes are induced by the radiation on the spectral distance between the two Bragg peaks. The two pink vertical lines indicate the start and stop of the irradiation run. Data extracted from [73].

**Figure 19 sensors-22-08175-f019:**
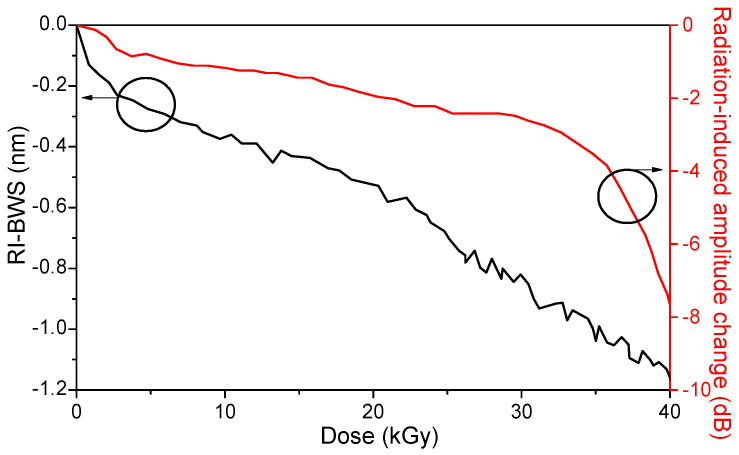
RI-BWS on the left (black curve) and peak amplitude reduction on the right (red curve) as a function of the dose, for an FBG written with an fs-laser in a CYTOP, under γ-rays. Data extracted from [38].

**Table 1 sensors-22-08175-t001:** Main characteristics of radiation environments (extracted from [14]) and list of references where the FBGs were tested in real applications.

Environment		Radiation Nature	Dose	Dose-Rate	Temperature	References
Nuclear Reactor Core		γ-rays neutrons	GGy 10^20^ n∙cm^−2^	<10^15^ n∙cm^−2^∙s^−1^	RT → 800 °C	[18,19,20,21,22,23,24,25,26,27,28,29,30,31]
Fusion-devoted facilities	Tokamak (e.g., ITER)	γ-rays 14 MeV neutrons	<10 MGy <10^18^ n∙cm^−2^	1 kGy/h <10^14^ n∙cm^−2^∙s^−1^	RT → 400 °C	[32]
	LMJ, NIF	X-rays γ-rays 14 MeV neutrons	<1 kGy	>MGy/s	RT	
High-energy physics facilities	LHC	photons electrons other particles	<100 kGy	<0.1 Gy/h	RT	[33,34,35,36]
Nuclear Waste Storage		γ-rays	<10 MGy	<10 Gy/h	RT → 90 °C	[37,38]
Medicine		X-rays protons	10^−2^ Gy → 50 Gy	<1 Gy/s	RT	[39]
Space		X-rays γ-rays protons electrons	<10 kGy	10^−5^ → 10^−3^ Gy/h	−200 °C → 300 °C	[40]

**Table 2 sensors-22-08175-t002:** Fiber Bragg grating classification with the inscription conditions, their origins, and their temperature operating range.

Type	Inscription	Origins	Temperature Resistance	References
I	UV (continuous or ns-pulsed) laser in photosensitive fibers, such as a Ge-doped one; OR fs-pulsed lasers (generally IR) in all fiber types.	Color centers ^1^. In the case of fs-pulsed IR laser, the defects are induced by multi-photon absorption processes.	T < 500 °C	[3,53,55]
II	fs-laser, whose power is higher than the damage threshold (~4 × 10^13^ W/cm^2^ for silica-based fibers), through phase mask or point by point	Densification and nano-structuration	T > 800 °C	[53]
Regenerated (R)	Seed FBG: type I grating in a photosensitive fiber, H_2_-loaded before or after the inscription; High-temperature treatment (T > 650 °C) that will erase the seed FBG before the appearance of the R-FBG.	Cristobalite, a crystalline polymorphic silica, generated by the high temperature and high pressure due to the hydrogen presence at the core/cladding interface (still debated).	T > 1000 °C	[56,57,58]
III or voids	fs-laser (whose power is higher than 10^14^ W/cm^2^) with the point-by-point technique.	Micro-voids surrounded by a shell of densified silica.	T > 1000 °C	[59,60]

^1^ It is worth noticing that, according some authors, type I FBGs can be due, along with color centers, to structural changes and densification [54]; however, the latter should not be erased by thermal treatments at temperatures lower than 500 °C. So, the subject is still controversial.

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
