# Peer review of "Radiation Effects on Fiber Bragg Gratings: Vulnerability and Hardening Studies"

_sensors, 2022, doi:10.3390/s22218175_

Round 1

Reviewer 1 Report

- This manuscript reports a comprehensive review of radiation effects on FBG including operating principle, inscription techniques, FGB classification, radiation effects on optical fibers/FBGs/exotic FBGs/FBGs in exotic fibers, and radiation-resistant FBGs.

- As far as I know, the manuscript successfully provided a guide on how the radiation affects the FBG performance according to different grating types under different parameters of the environments and different FBG fabrication/treatment techniques.

- The authors discussed about the best radiation-resistant FBGs with some recommendations. If possible, based on the review and experience, discuss about the idea how to design and fabricate the best radiation-resistant FBG, which can be different from type I, type II, regenerated, and type III, as it can be very helpful to fellow researchers in this field.

Author Response

Thanks for your positive comments.

Concerning the last point of the reviewer, we cannot implement this for all the FBG types. Type I and the regenerated ones do not work well under radiations. Moreover, whereas for type II a recipe exists and is reported in the manuscript, for the type III the good recipe seems to exist but it is not reported in literature.

Reviewer 2 Report

The paper is well organized with extensive literature review.

Author Response

Thanks for your very positive comments.

Reviewer 3 Report

A number of state-of the-art radiation effects of FBG based on different inscribing techniques had been reviewed. Several recent breakthroughs that influence the radiation vulnerability and hardness of diverse grating types are compared. Besides, to explain how radiation affects the performances of the FBG in real applications involving harsh environments, the existing FBG techniques and their potential in radiation-rich environment is concluded. Please adjust the figure resolution in this review paper to be consistent. And it would be very interesting for readers who study in FBG sensing system. I would recommend it to be published in this journal.

Author Response

Thanks for the positive comments.

We tried to improve the quality of the figures.

Reviewer 4 Report

In this paper, the effects of radiation on FBG are reviewed. From the fabrication of FBG to the influence of radiation on FBG, the mechanism is explained in detail. It is a review document worth reading for those who are interested in this research direction. Here are some suggestions:

1. Table 1 can be placed in the "Radiation Effects on Optical Fibers" section to improve the consistency of the paper.

2. The classification of FBG radiation response is relatively unfounded, and the order of each part can be properly arranged to strengthen the logic of the paper.

3. "Radiation Effects on exotic FBGs or on FBGs in exotic fibers" can be divided into two parts, and relevant content can be added. For example, special gratings can introduce tilted fiber Bragg Grating(TFBG) and chirped fiber Bragg grating(CFBG), and for exotic fiber parts, POF fabricated from different polymer materials can be introduced. Such as http://dx.doi.org/10.1016/j.yofte.2021.102593http://dx.doi.org/10.339/photonics6020036,etc

4. The conclusions part is not sufficient, it is suggested to detail the conclusions and give some analysis. For example, after the analysis of the above parts, put forward some suggestions on the fabrication of radiation-sensitive/insensitive FBGs and the future prospects for the application of FBG in this direction.

Author Response

The answers point by point are in the file here attached.

Reviewer 5 Report

This manuscript is very well organized and written. Radiation effects on FBGs are comprehensively presented, which is important for researchers in the field of radiation and FBGs. To make it more readable, I have minor suggestions.

1. "vulnerability and hardening" are not very accurate to describe the whole content of this manuscript. 

2. "3. Inscription Techniques", this section seems not very close to radiation effect, which could be compressed. 

3. In line 213, "n.cm", what does the "n" mean?

4. For Figure 3, I suppose the RIA is not related to "Refractive index change and grating period variation." 

5. In this manuscrdipt, there are many abbreviations. Some of them are quite similar, and could confuse the readers. The authors could reduce the number of the abbreviations.

6. in line 337, the "latter" should be replaced by the "fommer"? I suppose the cladding can't influence the grating under radiation.

7. "After irradiation, as for the RIA, a reduction of the RI -BWS is observed, known as “recovery”." This sentence is difficult to understand.

8. In line 806, DFB is normaly used for laser, it is not good to decribe a grating.

Author Response

The answers point by point are in the file here attached
